# Challenges and Outlines of Steelmaking toward the Year 2030 and Beyond—Indian Perspective

**Sethu Prasanth Shanmugam** [1], **Viswanathan N. Nurni** [2,*], **Sambandam Manjini** [1], **Sanjay Chandra** [2] **and Lauri E. K. Holappa** [3]

1   Research and Development, JSW Steel Limited, Salem 636453, India; ssprasanth47@gmail.com (S.P.S.); manjini.sambandam@jsw.in (S.M.)
2   Centre of Excellence in Steel Technology (CoEST), Department of Metallurgical Engineering and Materials Science, Indian Institute of Technology Bombay, Mumbai 400076, India; sanjaychandra@iitb.ac.in
3   Department of Chemical and Metallurgical Engineering, School of Chemical Engineering Aalto University, 02150 Espoo, Finland; lauri.holappa@aalto.fi
*   Correspondence: vichu@iitb.ac.in; Tel.: +91-22-25767611

**Abstract:** In FY-20, India's steel production was 109 MT, and it is the second-largest steel producer on the planet, after China. India's per capita consumption of steel was around 75 kg, which has risen from 59 kg in FY-14. Despite the increase in consumption, it is much lower than the average global consumption of 230 kg. The per capita consumption of steel is one of the strongest indicators of economic development across the nation. Thus, India has an ambitious plan of increasing steel production to around 250 MT and per capita consumption to around 160 kg by the year 2030. Steel manufacturers in India can be classified based on production routes as (a) oxygen route (BF/BOF route) and (b) electric route (electric arc furnace and induction furnace). One of the major issues for manufacturers of both routes is the availability of raw materials such as iron ore, direct reduced iron (DRI), and scrap. To achieve the level of 250 MT, steel manufacturers have to focus on improving the current process and product scenario as well as on research and development activities. The challenge to stop global warming has forced the global steel industry to strongly cut its $CO_2$ emissions. In the case of India, this target will be extremely difficult by ruling in the production duplication planned by the year 2030. This work focuses on the recent developments of various processes and challenges associated with them. Possibilities and opportunities for improving the current processes such as top gas recycling, increasing pulverized coal injection, and hydrogenation as well as the implementation of new processes such as HIsarna and other $CO_2$-lean iron production technologies are discussed. In addition, the eventual transition to hydrogen ironmaking and "green" electricity in smelting are considered. By fast-acting improvements in current facilities and brave investments in new carbon-lean technologies, the $CO_2$ emissions of the Indian steel industry can peak and turn downward toward carbon-neutral production.

**Keywords:** iron ore; coking coal; DRI; scrap; blue dust; natural gas; energy saving; decarbonization

## 1. Introduction

Steel manufacturing is a technologically complex industry having subsequent linkages in terms of material flow and plays a vital role in determining infrastructure and the overall development of a country. The global steel industry and its supply chain constitute 40 million jobs across the world. In 2019, India established itself as the second-largest steel producer with 111.3 million tons [1], constituting 5.9% of total crude steel production on the planet for the respective year, and it has ambitious plans to produce 250 million tons by 2030 with a per capita consumption aim of 160 kg. This is understandable, as the current per capita steel consumption in India is only 74 kg, which is much lower than the average global consumption of 229 kg [2]. India's crude steel production and per capita consumption with projection for the year 2030–2031 is given in Figures 1 and 2. Moreover,

the steel industries in India have challenges in terms of the non-availability of metallurgical grade coal within the country, leaner grades of ore, solid waste utilization, and water scarcity. Environmentally, the steel sector also faces challenges in terms of bringing down the carbon emission by 20% of 2005 levels within 2030. Indian steel manufacturers and associated metallurgical and mining industries are taking various measures and also need to take various measures in the future in terms of process improvements and developments to cater to the above needs. This paper attempts to elaborate on these challenges and speculate as to how the steel industries shall evolve in India in 2030 and beyond.

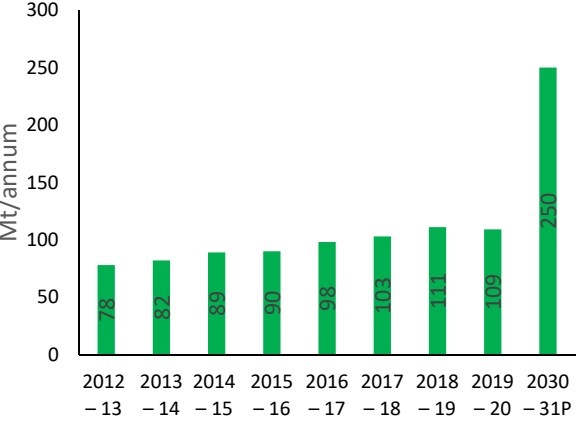

**Figure 1.** Crude steel production; (Graph plotted using data from ministry of steel [3]).

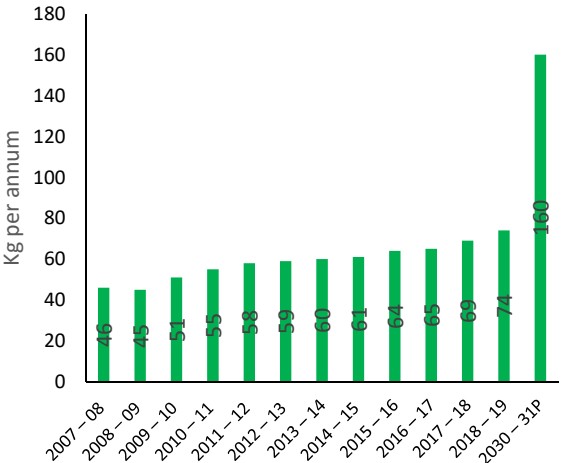

**Figure 2.** Per capita consumption of steel in India; Graph plotted using data from India Brand Equity Foundation [4]).

## 2. Organization of Indian Steel Sector

Based on production routes, Indian steel industries are divided into two types: industries producing steel primarily through a hot metal route using large integrated steel manufacturing units and others through primarily scrap and DRI in relatively smaller manufacturing units. The former ones are denoted as oxygen route manufacturers and the latter are denoted as electric route manufacturers henceforth in this paper.

Operations of the oxygen route manufacturers start from coke making, reducing iron ore primarily through blast furnace (BF) for the production of hot metal, and subsequently producing crude steel with standard specifications. There are also steel plants that use direct reduction (MIDREX), smelting reduction (COREX), and BF combination followed by the converter-arcing (CONARC) process. They all are ore-based production routes. Typically, these integrated manufacturers produce more than 5 million ton per annum from a single work location.

Electric route manufacturers primarily use scrap (they also produce DRI through a rotary kiln process), direct reduced iron (DRI), or hot briquetted iron (HBI) as raw material for the induction furnace route, and a few of them use the electric arc furnace route (EAF). Apart from these two, there are also independent hot and cold rolling units as well as sponge iron producers who use rotary kiln processes and pig iron producers who use mini blast furnaces. Typically, these manufacturers produce less than 1 million tons per annum from a single unit. Almost half of the Indian steel production is made from the electric route steel producers. Detailed process maps followed by Indian manufacturers are shown in Figure 3. The production capacity of various steelmaking processes in India is shown in Table 1. Crude steel production by process with its share percentage in Indian steel industry over the years is shown in Figure 4.

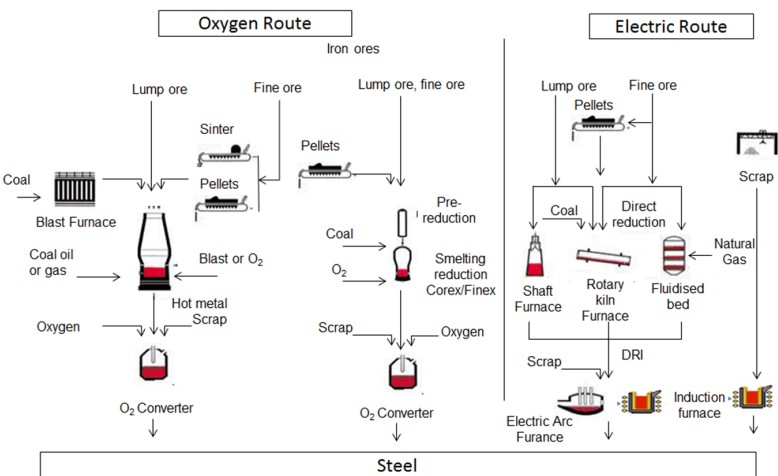

**Figure 3.** Indian steelmaking routes process map.

**Table 1.** Capacity of iron and steel industry in India as of 2018–2019; Source: The Energy and Resources Institute based on data from Joint Plant Committee.

| Segment | No of Units | Capacity (Million Ton) | Production in 2018–2019 |
|---|---|---|---|
| Blast Furnace | 59 | 80 | 72.5 |
| COREX | 2 | 1.65 | 1.4 |
| Sponge Iron | 318 | 49 | 37 |
| BOF | 17 | 57 | 49 |
| IF | 1174 | 49 | 33 |
| EAF | 56 | 42 | 28 |

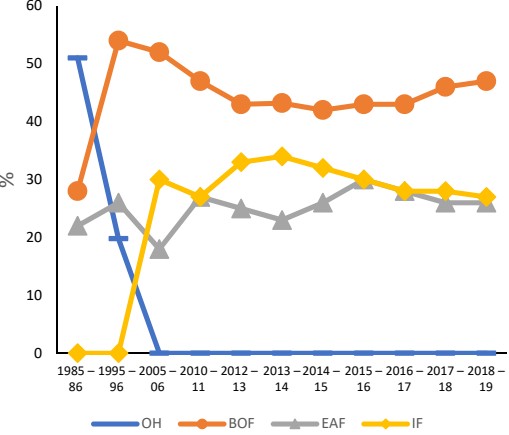

**Figure 4.** Crude steel production by process; share (%) OH—Open Hearth Process; BOF—Basic Oxygen Furnace; EAF—Electric Arc Furnace; IF—Induction Furnace. Graph plotted using data from India Brand Equity Foundation [4].

## 3. Current Trends and Challenges

### 3.1. Raw Materials

India is the 4th largest iron ore producer and the 3rd largest coal producer in the world. Coal is also identified as one of the major sectors of "Make in India", which is an initiative by the Government of India launched by the prime minister [5]. India is also the world's largest producer of sponge iron: about 37 million tons per annum.

### 3.1.1. Iron Ore

Proven hematite resources in India are around 29 billion ton, of which only 13% is high grade (>65% Fe), 47% are medium grade (62–65% Fe), and the remaining are low-grade ores [6]. During mining, lumps (−40 + 10 mm), fines (−10 + 0.15 mm), and slimes (<0.15 mm) are generated. For the efficient use of all these size ranges of ores, suitable ore agglomeration techniques are followed across the country. Over a while, sinter and pellet processes have improved and gained a significant place. Ore fines, −6 to 100 mm mesh size are being used for making sinter, and below 100 mesh fines are used for manufacturing pellets.

Indian iron ores have a major problem with high alumina and phosphorus. Highly friable hematite ore generates a huge quantity of fines during mining and crushing, which are rich in $Al_2O_3$. These fines are generally in the form of goethite (hydrated iron oxide) in which alumina is present in the matrix. Alumina exists in the form of gibbsite and kaolinite [7]. The high alumina content in these fines make them less amenable to physical separation and creates a problem in getting the final concentrate grade. In addition to magnetic and gravity separation, there is a need to explore the possibility of efficient beneficiation through froth flotation and selective dispersion with chemical aids. Some research efforts in this direction are being conducted at the National Metallurgical Laboratory, Jamshedpur, and Institute of Minerals and Materials Technology, Bhubaneswar.

As a result of the highly reducing atmosphere at the ironmaking stage, all the phosphorus in iron ore ends up in hot metal. Furthermore, if phosphorous is not removed in subsequent processes, it causes cold brittleness in steel. Efforts to remove phosphorous during mineral processing techniques are being explored using techniques such as thermal treatment, bioleaching, froth flotation, etc. [8]. The high P content in hot metal results in high P-containing slag in the steelmaking process and further poses difficulty in recycling this steelmaking slag back into the ironmaking process.

Blue dust is fine, powdery, soft, and friable ore rich in Fe content (65–67%) present in mines. Due to its fineness, it is not used directly in the furnaces. An estimated reserve of around 550 million ton of blue dust is available in India. India needs to find an appropriate technology to utilize this high-grade fine iron ore.

### 3.1.2. Coal

Although India is the third-largest coal producer after China and the USA, coking coal is only 15% out of the total coal mined [9]. In addition, Indian coking coal is not suitable for blast furnace due to its high ash content and hence requires extensive crushing and washing. About 85% of the coking coal requirements of Indian steel industries are met through imports [10]. Nevertheless, it invariably increases the coke rate in Indian blast furnaces compared to other countries. The cost of coke in the international market is expected to remain high in the foreseeable future and has forced the Indian manufacturers to go toward decreasing coke consumption by injecting pulverized coal. Increased usage of these injectants also paves the way for high production rates resulting from more oxygen input but decreases permeability, resulting in a larger pressure drop in blast furnaces [11]. Unfortunately, coal grades available in India are also found to be mostly not suitable for pulverized coal injection due to its high ash content. Thus, both for coke making as well as pulverized coal injection, Indian steel industries rely primarily on imported coal. Manufacturers in the electric route who use a rotary kiln process for the production of DRI/sponge iron also use primarily imported coal for their processes.

In summary, India has almost no reserves of coal that can be used in blast furnaces as well as other DRI-making units. This is one of the major issues that will shape the future of Indian steel industries and also should encourage efforts to decarbonize the Indian steel industry.

### 3.1.3. Alternative Fuels

Natural gas is the cleanest of fossil fuels, causing smallest $CO_2$ emissions. It is available in oil and gas fields located at the Hazira basin, Assam, Tripura, and Mumbai offshore regions [12]. A total of 90 Million Metric Standard Cubic Meters Per Day (MMSCMDs) of domestic gas production was achieved in FY 2018–2019. Apart from this, India imports Liquified Natural Gas (LNG) through six operational LNG regasification terminals with a combined capacity of $\approx$140 MMSCMD. Utilizing hydrogen-rich tuyere injectant such as natural gas helps to decrease the coke quantity as well as $CO_2$ emission of the blast furnace.

Other hydrocarbons such as waste plastics when heated up in the absence of oxygen, produce CO and hydrogen, which can be utilized in blast furnaces. However, before the injection of plastics through tuyeres, it needs to be crushed and pelletized (if necessary). Not all types of plastics can be injected; they should be segregated and sized. Indian manufacturers have yet to explore these advantages from plastics. Plastics can also be used in a coke oven as chemical feedstock [13].

### 3.1.4. Scrap

Major raw materials for electric route manufacturers are steel scrap and DRI. The import quantity of scrap and DRI production in India are shown in Figures 5 and 6. Domestic scrap supply from unorganized industry is around 25 MT (around 78% of the demand) in the year 2017–2018. [14]. However, the gap is expected to be reduced due to the increased production foreseen in the future, and it is likely to increase the scrap quantity considerably. In addition, the national scrap policy of 2017 envisages setting up a system that improves the processing and recycling of ferrous scraps through organized and scientific metal scrapping centers to minimize imports as well as to become self-sufficient in availability. In recent years, there has been an increasing trend in manufacturing steel through the scrap route, as it reduces greenhouse gas (GHG) emissions. Every ton of scrap utilized for steel production avoids the emission of 1.5 tons of $CO_2$ and the consumption of 1.4 tons of iron ore, 740 kg of coal, and 120 kg of limestone [15]. As of March 2019, 47 electric arc furnaces and 1128 induction furnaces are in operation across the country, and their main source of raw material is scrap.

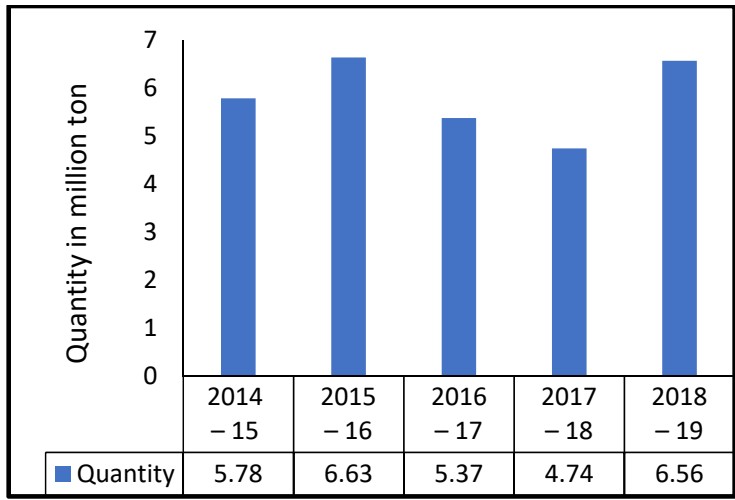

**Figure 5.** Steel scrap import over the years(Graph plotted using data from: Ministry of Steel).

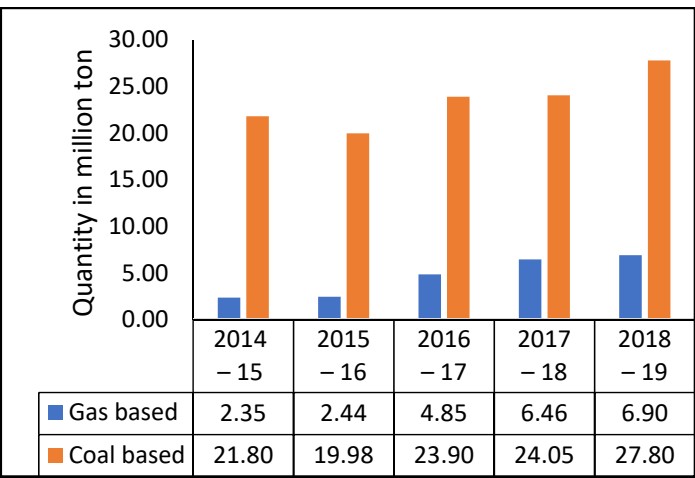

**Figure 6.** DRI Production (Graph plotted using data from: Ministry of Steel).

*3.2. Processes*

From the viewpoint of blast furnace operation, sinters and pellets in burden material give better bed permeability, decreasing the thickness of the cohesive zone due to narrower softening and melting ranges compared to lump ore. They also save energy since the calcination of limestone is avoided by making fluxed pellets/sinters, and we reduce alkali content in the blast furnace. As a result of these advantages, the use of lump ore is drastically reduced to around 10% [16].

The high alumina content in the iron ore poses a major challenge in blast furnace operation. To maintain appropriate slag viscosity, plants practice the addition of quartz and correspondingly the addition of lime to take care of basicity [17]. This results in a significant increase in the slag rate for Indian blast furnaces (375–420 kg/ton of hot metal as compared to 350–375 kg/ton of hot metal in China). High slag rates demanding more heat also result in higher coke rates. It also poses challenges toward increased pulverized coal injection in Indian blast furnaces.

High phosphorus in ore directly results in higher P% in the hot metal. To remove P, an excessive quantity of flux is necessitated in the converter process, resulting in significant temperature loss. If an external pre-treatment for hot metal dephosphorization is used, the process should be fast enough to match the BOF sequence.

Rotary kiln processes employ non-coking coal for converting iron ore to metallic iron and have the advantage of low investment cost. The energy balance of rotary kiln with theoretical and actual consumption showed only about 55% energy efficiency. A significant factor is that the waste gas liberated during the process remains unutilized at the moment [18]. It decreases the efficiency of the rotary kiln process. In addition, the utilization of this waste gas by waste heat recovery boiler experiences difficulty due to the accretion formation in the kiln and dust particles in the gas. These conditions make the rotary kiln highly inefficient compared to gas-based processes.

Typically, electric route manufacturers in India work with capacity less than 1 million tons per annum with the DRI-EAF/Induction Furnace route. Over the years, such small units have evolved closer to the market. It is also interesting to note that the electric route industries that are located in the western parts of the country use a higher proportion of scrap due to the accessibility of imported scrap by the sea route. On the other hand, on the eastern parts where ore reserves are abundant, they use rotary kiln units in combination with an induction or an electric arc furnace. Among electric arc furnace and induction furnace routes, the induction furnace is preferred because of its high yield (95–96% for IF and 92–93% for EAF), better electrical efficiency, and low investment cost [19,20]. The IF also has the characteristic of not using other forms of energy than electricity ($O_2$, fuel, carbon injection). However, unlike electric arc furnaces, at present, the maximum induction furnace capacity available in India is 50 tons only.

Although gas-based DRI is an environmentally effective process, the final carbon in the DRI is high compared to coal-based DRI from Indian plants (gas based 1.5–1.8%, coal based 0.2–0.25%). Thus, it makes it difficult for electric route manufacturers to reduce carbon in the product to the required range, especially when they use Induction Furnaces with acidic lining. However, manufacturers who use EAF prefer DRI from gas-based units, as the higher carbon content gives higher energy efficiency [21]. In addition, the high porosity, the low thermal, and the electrical conductivity of DRI are some of the other problems faced by electric route manufacturers. The proper selection of scrap and separation is much needed to have control over tramp elements in the final steel. Since most of the electric route manufacturers do not have extensive ladle-refining facilities, control over the nitrogen, hydrogen, and total oxygen contents are not present. These shortcomings are the reason why most of the induction-based manufacturers are making rebars and construction-quality steel rather than engineering and automotive-grade steel. In addition, the furnace lining of induction-based electric route industries is generally silica-based, and it results in ineffective dephosphorization compared to oxygen route products. The typical furnace lining life is 18 to 20 heats.

As the production is increasing, the demand for making clean steel is also increasing. It is the steelmaker's responsibility to fulfill the stringent quality requirements of the end-user. For example, the fatigue life of bearing steel greatly depends on the non-metallic inclusions. Bearing customers would want to have an inclusion size as low as possible (hard aluminum oxide and other oxide inclusions larger than 30 μm should be avoided). In that regard, reducing the total oxygen content in the steel becomes a necessary criterion. Reducing the total oxygen level from 15 ppm to less than 10 ppm would reduce the non-metallic inclusions as well. Achieving the low level of oxygen requires stringent control of the secondary steelmaking practice.

Steel manufacturers with large capacity and that have blast furnaces produce a versatile range of steel grades right from construction-quality steel grades to automobile, defense, railways, and aerospace grades. Larger capacity manufacturers have their rolling mills inbuilt in their facilities, and they produce a plethora of products in the form of bars, flats, wire rods, hexagon bars, etc.

Steel manufacturers with induction and arc furnaces produce mostly construction-quality grades in the form of rebars due to the factors mentioned earlier. Some manufacturers have a vacuum degassing facility and argon oxygen decarburization facility (AOD) and produce high-quality alloy steels for automotive applications and stainless steel grades with a smaller heat size. In Indian steel market conditions, highly critical steel grades for applications in space and aerospace equipment are imported. India needs to grow in terms of high-value niche special steels.

*3.3. Environment and Energy*

Steel industries generate around 30% of the total $CO_2$ emissions from all industrial sectors on the planet. The reason for this huge quantity is due to the usage of coke and the high consumption of energy factory-wide. Nearly 65% of the emissions of oxygen route industries come from ironmaking processes, out of which 90% is contributed by coke and coal. Steel industries all around the world are striving hard to improve the energy efficiency of blast furnaces by approaching the theoretical limits of production and carbon consumption.

The iron and steel industry is one of the major energy consumers in India as well. By using the best available technologies (BAT), the specific energy consumptions are BF-BOF route-16.4 GJ/TCS (ton of crude steel), COREX-BOF-19.3 GJ/TCS, DRI-EAF (coal-based) route-19 GJ/TCS, and DRI-EAF (gas-based)-15.9 GJ/TCS [14]. However, major steel plants in India have a specific energy consumption of 27.3 GJ/TCS [14]. Although there exists a substantial potential to save energy by adopting the best practices and newest innovations for reducing energy consumption, reaching this target can be quite challenging considering the quality of raw materials in India. Thus, the authors opine that arriving at a target-

specific energy consumption considering the local raw material quality can be quite fruitful in defining the road map for steel technology for India.

A major challenge for the electric route with arc furnaces or induction furnaces is the electrical energy consumption. The melting process starts from room temperature, and there is no heat recovery from off-gases. Typically, Indian manufacturers have electricity consumption 600 kWh/T compared to the world average of 416 kWh/T. However, DRI-EAF based routes generally consume less energy than companies that use EAF alone. The specific energy consumption of gas-based DRI is 10.46 to 14.43 GJ/T, and coal-based plants have 15.9 to 20.9 GJ/T [22].

Waste disposal and treatment is a big challenge for steel manufacturers. The steel industry generates solid, liquid, and gaseous wastes, and the most common are iron and steel slag, scrap, sludge, effluent, flue gases, etc. Approximately 0.4 to 0.8 tons of solid waste is generated for a ton of liquid steel (also, 0.5 tons of effluent water and 8 tons of moist laden gases) [22]. The waste management systems practiced by steel industries involve processing the wastes for a recycling in-plant process or disposing by appropriate methods. Ironmaking slag such as BF or COREX slag is used as a raw material for slag cement making and for producing slag sand, which can replace river sand. Due to the presence of free lime, BOF slag is not used in construction applications and generally goes for landfilling. Steel plants are developing and adopting methods for accelerated weathering of the steel slag and use it in construction activities, tide breakers in the coastal areas, and soil ameliorants. Some of the other applications for weathered BOF slag are rail ballast, cement making, road and paver blocks, and brick and tiles manufacturing [23].

On the contrary, electric arc slag is granulated and used as agglomerates in road construction and similar applications. Fine granulated EAF slag is also used in shot blasting and industrial water filter applications. Ladle-refining slags have high CaO, MgO, and $SiO_2$ contents and could be used as a raw material in cement production [24]. However, the powdery nature of the LF slag makes it difficult to handle. In addition, the presence of high sulfur, heavy metals such as Cr, V should be avoided and thus necessitates proper screening and care. One of the important solid wastes to be recycled is the slag from BOF, as it cannot be dumped due to its high phosphorus content.

## 4. Possibilities and Opportunities

### 4.1. Raw Materials

a.    Iron Ore

Demand for iron ore has been increasing with the increased production of iron and steel in developing countries such as India and China. However, the quality of iron ore has deteriorated over the years globally due to long-term mining. The low-grade iron requires beneficiation before agglomerating for use in the iron-making process. The iron ore interlocked with silica and alumina has to be liberated for efficient beneficiation [25]. This requires finer communition, resulting in iron ore fines often finer than 300 microns. It means extra energy consumption, too. The pelletization process is better suited as an agglomeration process for finer fraction iron ores. It is envisaged that the use of pellets in the Indian blast furnaces will increase the replacement of the usage of sinter. For pellet induration, natural gas can be used, which is less polluting than the coke fines used in sinter making.

A large quantity of fine iron ore in the form of slime arises after beneficiation having high $Al_2O_3$, making it unsuitable for blast furnace operation. Slime beneficiation techniques such as hydro-cyclone and magnetic separation techniques are being explored to produce an iron concentrate with ~63% of Fe [25]. Bhagwan Singh et al. from NMDC utilized the blue dust concentrate to produce ultra-pure and high-quality ferric oxide, premium-grade sponge iron powder using hydrogen as a reducing agent [26]. This kind of process is expected in the near future for full-scale production.

b.    Coal substitutes

Plastics can be used as chemical feedstock in the coke oven and in blast furnaces. Waste plastics contain carbon and hydrogen, and if it is thermally heated along with coal in a coke oven, it generates coke, coke oven gas, and hydrocarbon oil. In the latter case, the plastics shall be dechlorinated, pulverized, and injected through the tuyeres of the blast furnace as a reducing agent. Nomura et al. have analyzed the different types of plastics used in the coke oven and out of which only PE (polyethylene) and PVC (polyvinyl chloride)-type plastics have a small effect on properties, and particularly, PE increased the strength of coke. In contrast, PS (polystyrene) and PET (polyethylene terephthalate) inhibited the strength [13]. The size of the granules also requires optimization for different varieties of plastics. India can explore the possibility of using waste plastics in coke ovens, as it ensures a sustainable social system. For every ton of plastic used, coke can be reduced by 750 kg.

c.    Alternate Fuels

India is expecting 15% of the energy mix from natural gas utilization by 2030 from 6% today. Currently, India has six operating LNG terminals. India started building an LNG port at Kakinada by 2019, and the government has plans to build 11 more terminals, particularly on the east coast. Apart from this, India is working on constructing a national gas grid, increasing to around 14,000 km of additional pipeline networks across the country. These plans would increase the availability of natural gas for steel manufacturers. With the increased availability of natural gas, the projections for sponge iron production by the process route in India by 2030 is given in Figure 7. Although the share of the gas-based production will increase from 25% to 33%, also the coal-based production is planned to increase, thus ruling out the efforts to mitigate $CO_2$ emissions.

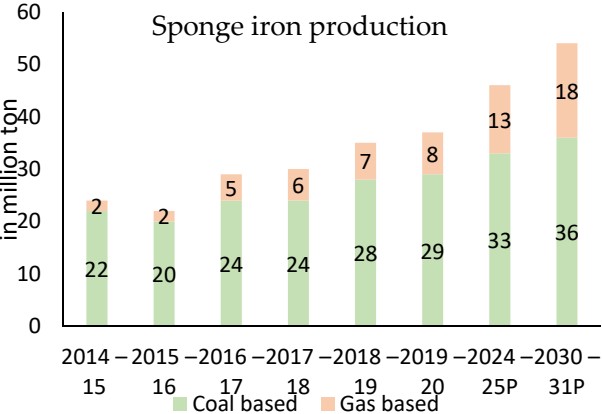

**Figure 7.** Sponge iron production forecast (Graph plotted using data from Ministry of Steel).

d.    Scrap

The Ministry of Steel has issued the steel scrap recycling policy [14]. It aims to promote the metal scrapping centers and also ensures the scrap processing and recycling from various sources. It envisages a structure to provide standard guidelines for the collection, dismantling, and shredding of scrap. The scrap requirement of India in 2030 is expected to more than double from the current level 32 MT/2019 to 70–80 MT [12]. The Indian government expects that the increased production of vehicles for the last two decades would generate a continuous flow of steel scrap for recycling to steel production. Once scrap becomes more available, then DRI + scrap utilization through an induction furnace or electric arc furnace may increase in the future.

With the increased availability of natural gas and scrap, the recent history and the projected process-wise crude steel production for the year 2030 is given in Figure 8. Principally, the trend is positive, but again, the projected strong production growth will inevitably lead to increasing emissions.

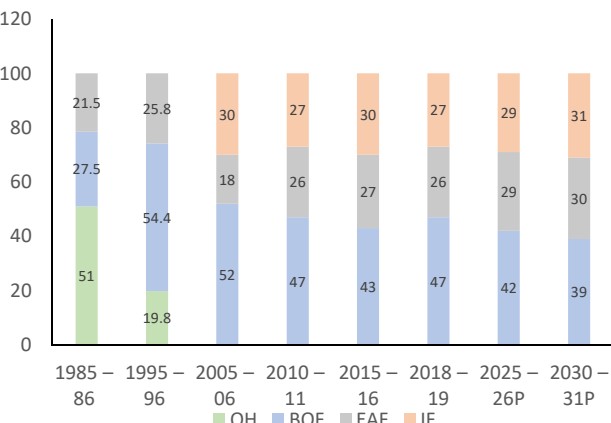

**Figure 8.** Process-wise crude steel production since 1985 and a forecast to 2030 (graph plotted using data from Ministry of Steel).

*4.2. Processes*

4.2.1. Improvements in Existing Processes

The problem of high alumina in ore can possibly be handled through appropriate modification in blast furnace design. High alumina resulting in high slag rates limits the production rates, especially because of the complex fluid dynamics in the bosh region. If the belly size is increased, the superficial gas velocity can be brought down, which can permit higher production rates.

Apart from the technologies discussed earlier (PCI, TGR-OBF), there are other methods aiming at reducing specific energy consumption and emissions such as coke dry quenching, top recovery turbine, and recycling of blast furnace slag, as well as the utilization of alternative fuels such as biomass, waste plastics, and natural gas. To implement waste heat recovery technologies such as the preheating of coal charge and coke dry quenching technology, the introduction of a waste heat recovery system is necessitated, which can bring down the requirement for fuel input in coke making from a level of 2.3 GJ/T of coal charge to 2.1 GJ/T of coal charge.

Pulverized coal injection (PCI) into blast furnaces is a common practice nowadays to improve the BF economy. For every ton of coal injected in the blast furnace, 0.85 to 0.95 tons of coke production can be avoided. To increase coal injection, one needs to adjust the parameters to achieve optimum permeability. Some ideas to achieve the same are (1) reduction of the size of coal particles—finer particles provide greater surface area and increase the rate of combustion—and (2) catalyzing the gasification reaction by treatment of coke particles with catalysts such as the slime of fine particles. Increasing the PCI is associated invariably with the pressure drop due to a decrease in coke fraction from the top. Some measures to avoid that are (1) exploring the possibilities of making coke layer thickness larger at the expense of the number of coke slits and (2) charging coarse particles of ore on the top of a coke layer [11].

Injection of plastics after proper collection, segregation and preprocessing, biomass, and even solid injectants such as flux and BOF slag into a blast furnace can be explored for improving the blast furnace production and reducing the carbon footprint. With increasing environmental awareness and the policies of the state, these efforts are expected to gain momentum.

Oxygen blast furnace with top gas recycling technique (TGR-OBF) is another process being explored for improved energy efficiency and reduced emissions to the environment [27]. This technique actually removes the carbon dioxide from top gas and the remaining gas, which is reducing in nature, is injected back into the blast furnace through tuyeres. Nogami et al. and Danloy et al. verified that heat demand will be sensibly reduced and productivity improved by top gas recycling [27,28]. N.B. Ballal reviewed this technique with the 0d model and found that top gas recycling would increase the CO partial pressure of bosh gas (from 35% to 42.7% with 20% recycle and 100 kg/ton of hot metal), resulting in

faster reduction kinetics [9]. No blast furnace is running with this technique, but India can explore this possibility. In addition, dust in the blast furnace gas can be separated and can be sent back to the furnace, especially if it is low in alkali oxides. Thus, it will increase the carbon efficiency in the BF as well as reduce coke consumption.

A study on rotary kiln energy efficiency by Nishant et al. [29] identified the possible areas where energy is lost in the form of waste gas, cooling of sponge iron, high exit temperature of clean waste gas, and intrusion of air. They suggested ways to reduce these losses by process integration principles. Ideas such as these should be promoted in the coal-based rotary kiln DRI manufacturing to make the process efficient. Otherwise, with the increase in the shift toward gas-based processes because of the plethora of advantages, rotary kiln operation would see an end in the future.

The recovery of sensible heat from metallurgical slags is a challenge. An array of energy recovery means from slags such as thermal, chemical, and thermoelectric generation technologies have been investigated. Among the technologies, thermal methods are developed the most, as it does not have complex multi-step technology as well as temperature constraints. In TATA steel (Sridhar et al. [30]), in the pilot scale, waste heat from BOF slag was utilized for splitting water molecules to harvest hydrogen ($H_2$) that is cleaner and available at low cost. Using 15 kg of slag, they were able to get gas that contains as much as 23% hydrogen. The main challenges reported were safe handling of the large quantity of slag.

In 2020, The Energy and Resources Institute (TERI) summarized the energy-efficiency potential in the Indian steel industry [31]. By modernizing equipment and processes and adopting best available technologies (BAT), energy consumption can be reduced by 25–30% from the current level. The potentials in different process stages in the BF-BOF route are shown in Figure 9.

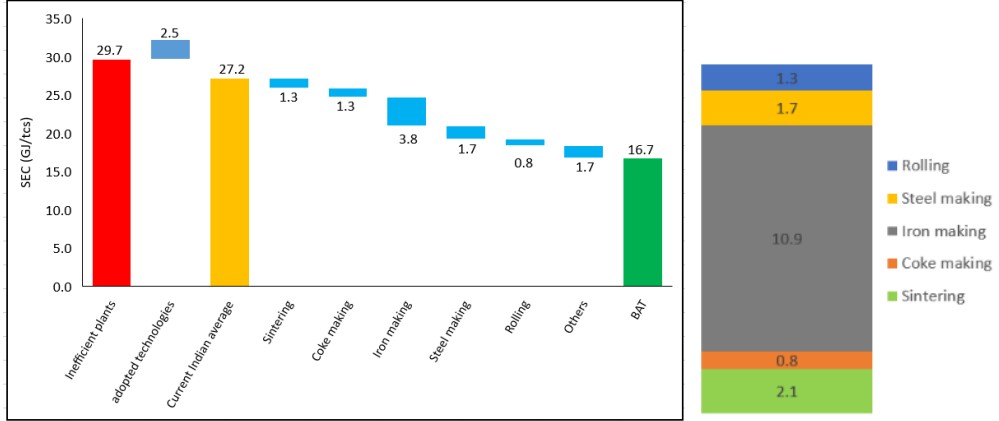

**Figure 9.** Specific Energy Consumption (SEC) and potentials for improved energy efficiency in BF-BOF route and breakdown of BAT (right) Reprinted with permission from TERI [31] Copyright 2020, The Energy and Resources Institute.

Due to the high phosphorus in Indian iron ore, high P in BF hot metal is inevitable. At the same time, the demand for low and ultra-low P% in high-quality steels is growing, which sets an extreme challenge to the converter operation. Dephosphorization can be done as pre-treatment of hot metal in torpedo ladle or at the basic oxygen furnace itself, depending on the phosphorous percentage in hot metal. Typically, lime or lime with fluorspar used as flux and Fe oxide with or without $O_2$ is used as an oxygen source for dephosphorization. It can be divided as low oxygen activity with a high basicity process or high oxygen activity with low basicity based on the slag composition. Liu et al. [32] found that the most favorable temperature for P removal is around 1300–1400 °C. The dephosphorization reaction is a slag–steel interface-controlled reaction. The distribution coefficient between slag and iron is temperature dependent, getting lower at high temperatures. On the other hand, a very low temperature under 1300 °C makes slag viscous and causes poor reaction kinetics. $CO_2$ injection has been applied in controlling the bath

temperature instead of sending oxygen alone. In addition, it also reacts with bath elements, generating additional CO gas and thus intensifying the stirring effect. When 15% $CO_2$ was mixed in $O_2$ blown through the top lance in a 300 ton converter, the dephosphorization rate increased from 56 to 63% [31]. As a drawback, the cooling effect of $CO_2$ injection results in a notable reduction in scrap-melting capacity. This idea of $CO_2$ injection could be explored in Indian steel industries, as it helps both improve the quality of steel and the environmental point of view, provided that the required $CO_2$ for injection is captured through CCS technology, which is explained in a later part of this paper.

In the electric furnace route, although utilizing gas-based DRI is beneficial to the industry, the increased carbon and phosphorous contents limit the usage of it. Double slag practice can be envisaged for the induction furnace route to have better control over the phosphorus.

### 4.2.2. New Technologies/Processes

The regular supply of natural gas to gas-based processes such as Midrex is a real concern. Therefore, some plants are going for the in-plant generation of synthesis gas from coal. Syngas or synthesis gas is a product of the gasification of carbon-containing fuel having a mixture of hydrogen, carbon monoxide, and carbon dioxide. Some plants using the COREX route produce hot metal as well as additional gas, which can be used for gas-based reduction, e.g., via a Midrex reactor [33]. The mixing of coke oven gas with synthesis gas for reduction is also practiced by some manufacturers, and efforts are being undertaken to build coal gasifiers to produce synthesis gas. These technologies can more or less reduce $CO_2$ emissions compared to the current industrial practices, but they are not able to compensate for the emissions due to the planned strong growth of the steel production in India.

Recently, the use of hydrogen in ironmaking to replace coal or natural gas has received growing attention. In the first step, instead of pure hydrogen, the usage of different blends of gases, from natural gas to syngases, is necessitated. There are three major gas-based processes available in the world—namely, Midrex, HYL, and Circored. The Midrex process already has been operated with a level of 55 to 75% hydrogen concentration.

The implementation of hydrogen in the steel industry will be strongly dependent on the decarbonization of the power sector. Depending on the production process and utilization of energy source, hydrogen generation can be called green, blue, gray, or pink. Green hydrogen comes from splitting water by electrolysis, gray hydrogen is when natural gas is split into hydrogen and $CO_2$ through various processes. Blue hydrogen is the same as gray except the $CO_2$ is captured through CCUS technologies. Pink hydrogen is the same as green except it uses nuclear energy for splitting water. Since India possesses a significant capability in the generation of renewable energy sources through both solar and wind, the electrolysis process can be utilized for the production of green hydrogen. However, the process is electrically as well as water-intensive and requires high capital cost. India is expanding its natural gas grid far and wide, so the opportunities for lowering industrial emission as well as hydrogen utilization would grow stronger.

ULCOS—The Ultra-Low Carbon Steel making program is a co-operative European research and development initiative that was launched in 2004 to search for the process that could in the future when fully developed establish the potential of large cuts in $CO_2$ emission for steel production from iron ore. Several concepts have been investigated in parallel, using modeling and laboratory approaches to examine the potential of processes in terms of $CO_2$ emissions, energy consumption, cost, and sustainability. HIsarna is one such process identified through the ULCOS program; it is a combination of Isarna and HIsmelt technologies to produce liquid hot metal directly from iron ore [34]. It requires neither agglomeration of iron ore nor coke. It also has the potential to utilize the non-coking coal reserves in India. It is efficient in energy, and it has a lower carbon footprint than conventional processes. It reduces energy consumption by at least 20% and $CO_2$ emission by 20% [34]. One benefit of this process is the production of a very pure stream of $CO_2$,

which can be collected cost-effectively. A pilot plant of HIsarna was constructed by Tata Steel Ijmuiden, and in November 2018, the company announced that a large-scale HIsarna will be constructed at Tata Steel, Jamshedpur [35]. It could be a path-breaking step for the steel industries in terms of energy consumption and environmental emissions. The capital costs of the HIsarna process are also 10–15% lower than the conventional BF-BOF route due to excluding sinter plant and coke ovens. It could also help in achieving a low phosphorus level, as it maintains a lower temperature in the bath.

Finex Technology—Considering the large amount of iron ore fines produced during mining as well as the abundance of non-coking coal (combined with limited resources of coking coal), Finex technology was considered to be one of the potential technologies for India [36]. In 2015, POSCO was planning to install a 12 MT steel plant using Finex technology [37]. However, because of non-technical reasons, the plan was discontinued. Considering the success of COREX technology in two plants in India, Finex technology is a viable route for the future of Indian steel sector.

Flash Ironmaking Technology (FIT) is a potential idea to utilize the huge quantity of blue dust existing in Indian mines. It is based on the reduction of fine iron ore particles to convert them directly to metallic iron with suitable reductants (such as hydrogen, natural gas, coal gas, or a combination of gases) [38]. This technology has been developed in the University of Utah as a part of American Iron and Steel Institute's $CO_2$ breakthrough program. Agglomeration techniques can be avoided in this technology, thus avoiding the usage of coke. Thus far, it has been tested only in bench-scale experiments. However, this idea can be explored in the future, as it helps in reducing energy consumption and $CO_2$ emission [39].

Biomass can be used as a renewable fossil fuel to mitigate the emission of $CO_2$. It can be charged at the top of a blast furnace along with coke, injected through tuyeres, or blended with coke to produce bio-coke. However, it is a challenge due to lower Coke Strength after Reaction (CSR) and higher Coke Reactivity Index (CRI) as compared to metallurgical coke. Biomass fuel shall be effective in reducing fossil $CO_2$ emission and more effective in the mitigation of $SO_x$ and $NO_x$ [40]. In Brazil, there are numerous mini blast furnaces based on charcoal [41]. Very few blast furnaces in the world tried using biomass. Stubble burning such as in Punjab and Haryana causes extreme air pollution every year. Instead, large amounts of biomass could be produced and potentially utilized by steel manufacturers. This possibility is yet to be explored for sustainable carbon footprint and its availability for the steel sector. The optimization of the biomass value chain and the efficient conversion technologies are of high importance for replacing fossil fuels in the near future.

### 4.2.3. Energy Efficiency in Indian Steel Industry

Significant improvements in energy efficiency can be achieved in different unit processes through improved operation of equipment via optimized integration of in-plant energy flows and by upgrading process equipment to commercially available Best Available Technology (BAT). An energy saving of around 25–30% per ton of crude steel can be achieved by improving the operational efficiency and adopting BAT for all the units of the BF-BOF production pathway, relative to the global average energy intensity for this route today (Figure 9 [31]). Using electricity to substitute for fossil fuels in the provision of process heat in equipment outside the main process units, particularly in preheaters and boilers, is another option where electrification makes a change.

### 4.3. Energy and Environment Conservation

Environmental emissions of the steel industries primarily relate to the air and water pollution and solid wastes [42]. The global concern over climate warming is attributed to the $CO_2$ emissions. An extensive decarbonization of the world steel industry until the middle of this century is a common target. India is liable to take care of its own share in this global crisis. In spite of the planned elevation in the steel production, the $CO_2$ emissions

should peak in the next years and then turn down toward carbon neutrality in the middle of the century or shortly after. That demands the right selections and epoch-making actions. The most essential ways to solve this dilemma are discussed in the following sections.

### 4.3.1. Carbon Sinks, Capture, and Storage

One of the best ways to capture $CO_2$ is the natural sequestration by plants. India aims to create a sink of 2.5–3 billion tons of $CO_2$ through additional forestation and tree cover by 2030. Apparently, through large-level community involvement, India has launched the Mahatma Gandhi National Rural Employment Guarantee Act (MGNREGA), which focuses on environment and natural resource conservation. A detailed review of its potential impact on carbon sequestration through the conservation of green natural resources has been carried out by the Indian Institute of Science, Bangalore [43]. Steel plants are also actively involved in such afforestation drives [44].

Transition to processes based on low-carbon and carbon-free clean energy is a key factor to conserve the energy and environment. During the transition period, Carbon Capture and Storage (CCS) affords further means to cut $CO_2$ emissions. With CCS technology, up to 90% of the $CO_2$ emitted can be captured, compressed at high pressure, converted into a liquid, and injected to geological formation sites to be stored underground without significant leaking for hundreds of years. The implementation of this process along with TGR-OBF, HIsarna, etc., can significantly reduce the emissions levels. One major concern for CCS deployment in India is to find an accurate and suitable geological site for the installation of CCS. Another issue is that CCS significantly increases the cost of electricity while reducing net power output, which is often cited as being the biggest barrier to the acceptability of CCS in India [45].

As mentioned earlier $CO_2$ can be injected into geological formation sites such as saline aquifers, basalt formations, depleted oil and gas fields, and non-minable coal seams to fixate $CO_2$ as carbonates. An initial geological study suggests that 500–1000 Gt of $CO_2$ can be potentially stored around the subcontinent [46]. More specific studies pertaining to India in this direction are needed, and furthermore, India is closely studying these developments in developed countries in terms of implementation in the actual scale and subsequently planning to adopt them.

The adoption of CCS in India, especially to thermal power plants, would create major challenges, and it may be offset by the following means [45]:

- Development of IGCC (Integrated Coal Gasification Combined Cycle) technology, which gasifies the coal and uses a combined cycle (combination of gas and steam turbines) to generate electricity;
- Indigenous development of capture and compression equipment for cost efficiency;
- Improved blending and beneficiation of coal;
- Membrane-based capturing.

CCUS combined with enhanced oil recovery (EOR) can be a win–win situation for India, since it can help arrest declining output from oil and gas fields. The technology plays an increasingly important role in achieving carbon neutrality. India is now exploring its CCS potential in the power sector. A plant at the industrial port of Tamil Nadu's Tuticorin has begun capturing $CO_2$ from its own coal-powered boiler and using it to make baking soda [45].

Interestingly, steel slags themselves can potentially be used for $CO_2$ capture. Among the industrial wastes, iron and steel slags have the maximum $CO_2$ sequestration potential [47]. Raghavendra et al. have given an excellent review of various ways in which $CO_2$ can be captured such as hot route carbonation by treating the hot slag with $CO_2$, direct route involving gas–solid, thin film, and aqueous slurry carbonation, and an indirect route involving pH and pressure swing $CO_2$ absorption techniques [48].

Indian steel plants have started looking at some of these options more closely to develop in-house technologies to bring down the carbon footprint [49].

The hydrogen-based DRI route (alongside blending of electrolytic hydrogen into current blast furnaces and DRI units) and integration of CCUS in various production units shall account for substantial shares of emission reduction.

### 4.3.2. Prospects of Power Sector in India

India's power sector is currently dominated by coal. Toward the objective of carbon-free energy, India has set itself a target of installed capacity of 175 GW from renewable energy sources by March 2022. To control the obnoxious emissions, more efficient coal-based units are being commissioned, and inefficient units are being discarded. The total installed capacity of power generation (i.e., maximum electric power output) in India is 363 GW, and the distribution of energy generation capacity across different sources and projection for 2029–2030 is given in Table 2 [50]. The renewable energy given includes solar, wind, and biomass. The shift toward renewable energy resources can be evidenced. The emission factor of the Indian grid electricity was roughly 900 g $CO_2$/kWh in 2019 [51]. The Energy and Resources Institute (TERI) in Delhi launched a comprehensive study on decarbonization pathways for the power sector, including also energy-intensive industries such as steel [30]. According to their chart, the grid emission factor might decrease to 550 in 2030 and to around 100 g $CO_2$/kWh in 2050. India is having an ambitious figure of 52% generation through renewable sources in 2030, it seems to be an elusive task.

**Table 2.** Distribution of electricity generation capacity 2019–2020 and projected distribution for 2029–2030 [50].

| Sector | Installed Capacity-2019–2020 (GW) | % | Estimated Capacity 2029–2030 | % |
|---|---|---|---|---|
| Hydro | 45.4 | 12.50 | 73.45 | 9.31 |
| Thermal–Coal + Lignite | 203.6 | 55.90 | 266.9 | 32.66 |
| Thermal–Gas | 24.9 | 6.90 | 25 | 3.07 |
| Thermal–Diesel | 0.5 | 0.10 | | - |
| Nuclear | 6.8 | 1.90 | 16.9 | 2.32 |
| Renewable energy | 82.6 | 22.70 | 450.1 | 52.63 |
| Total | 363 | | 831.5 | |

## 5. Emissions Mitigation in the Indian Steel Industry—Summarizing Discussion

Currently, the Indian steel sector can be broadly classified into (1) plants using oxygen route steelmaking processes and (2) plants using electric route steelmaking. Oxygen route steel makers mainly use blast furnaces, but additionally, there are two plants with four COREX units to produce hot metal for making steel. On the other hand, the electric route steel makers use DRI either produced in coal-based units or gas-based units and imported scrap for making steel. An Indian specialty is the huge number of small-size induction furnaces (the current number 1174) with an annual production of only tens of thousands of tons of steel each. The current ratio of the oxygen route versus the electric route, EAF and IF is approximately 45:30:25. Overall, the Indian steel production is expected to grow to 250 million tons per annum by 2030 and it is still growing, albeit retarding toward the year 2050 [31].

In blast furnaces, coal injection (PCI) up to 200 kg per ton of hot metal is already in use, and this is expected to only marginally increase due to constraints in operation. Additionally, steel plants are exploring possibilities of injecting gases having a significant proportion of hydrogen through tuyeres and/or in the stack to reduce the carbon footprint. The gas for the "step changes" injection may come either from imported natural gas or large gasifiers using domestic coals, and imported cheaper coals can also be envisaged. With the increasing usage of gases in these plants, DRI-based units will take a prominent role. Furthermore, if the hydrogen economy becomes a reality in the future, these plants can shift

toward hydrogen-based DRI production. Further owing to the virgin ore-based production and large capacity, the oxygen route plants continue to produce high-end flat products.

As of 2020, the utilization of hydrogen in the steel industry could play a growing role in mitigating $CO_2$ emissions as well as improving the existing process. The cost of hydrogen is expected to fall with the rise in the utilization of gas-based processes and increase in renewable energy sources. Under a future low-carbon situation, we can foresee a tremendous growth in demand for hydrogen from 2030. Thus, hydrogen production will increase rapidly from 2030 to 2050, enabling new capacity additions utilizing hydrogen-based reduction processes.

Plants that use electrical steelmaking routes, primarily based on induction furnaces, produce construction-quality steel grades and are located close to the markets, i.e., spread across the country. Many of these units use also DRI, which is primarily produced through coal-based processes. With increasing environmental threats, these plants will be forced to shift from coal-based to gas-based processes. At the same time, a transition to larger capacity units is reasonable for economic reasons. The gas for these DRI units may be imported natural gas or from coal gasification units. These units shall supply small electrical route steel makers located near consumer markets where the scrap also will be available in the future as the consumption and recycling of steel rise in the country. At present, induction furnaces with a maximum capacity of 50 tons are being used. Depending on the demand from the consumer markets, especially near large cities, electric arc furnace-based steelmaking units with large capacities can also emerge.

Incremental advancements can be achieved through improvements in the operation of equipment and by upgrading process equipment to commercially available Best Available Technology (BAT), which reduces the energy demand required per ton of process output. An energy saving of around 20% per ton of crude steel can be achieved by improving the operational efficiency and adopting the BAT for all the units of the BF-BOF production pathway, relative to the global average energy intensity for this route today [30]. Step changes in efficiency shall be achieved by switching to alternative production pathways such as electrification or other fuel shifts. Using electricity to substitute for fossil fuels in the provision of process heat in equipment outside the main process units, particularly in preheaters and boilers, is another option where electrification can make a big change.

For meeting the global environmental goals, India is expected to shift from coal-based processes to more gas-based processes. These changes are expected in both sectors: namely oxygen route steel makers as well as electric route steel makers. It will be facilitated by the reduced prices of LNG due to the increasing number of ports, thus helping gas find a larger foothold in the Indian mix. Increased the replacement of coal-based Direct Reduced Iron (DRI) with gas-based DRI will also help in attaining reductions in final energy consumption and $CO_2$ emissions. This shift in processes in the iron and steel sub-sector will result in a change of the industrial ecosystem in the country. By applying the necessary improvements in the process such as moving toward gas-based DRI, increased hydrogen usage and the implementation of new technologies such as the CCUS target on specific energy consumption and specific emissions shall be achieved. Figure 10 shows a scenario of emission reduction potential for the Indian iron and steel sector. It incorporates the incremental improvements discussed previously and shown in Figure 9 and, additionally, an introduction of HIsarna and a conservative transition to hydrogen [31]. The projected production growth would mean a tremendous rise in emissions from 2020 to 2050 (Baseline). The planned actions to cut emissions would result in a 56% reduction by 2050 to a level where the total emissions would be approximately 20% higher than today! The peaking year would be around 2040. These issues proceed from the predicted radical growth of the steel production. The concurrent enterprises to raise the production and to cut the emissions are surely contradictory.

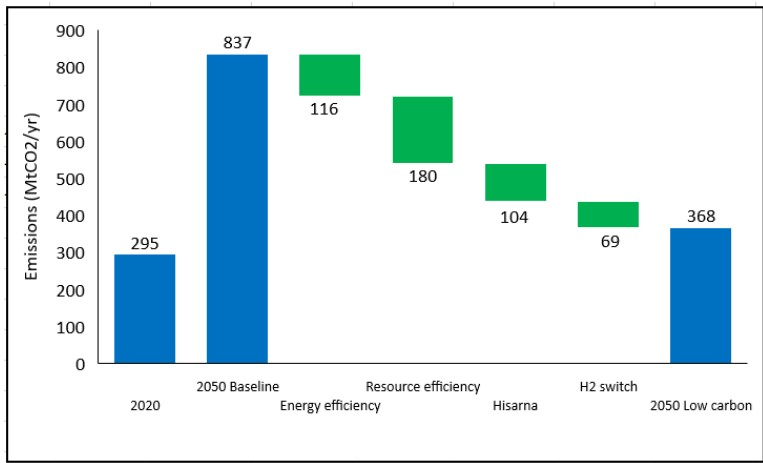

**Figure 10.** Emission reduction potential of the Indian iron and steel sector Reprinted with permission from TERI [31] Copyright 2020, The Energy and Resources Institute.

For fast-growing economies with a rising emissions trajectory, the need to understand the key variables that impact the choice of a peaking year is as critical as the determinants for the selection of a net-zero year. An analytical formulation [52] shows that the economic growth rate significantly impacts the 'effort gap'. For India, peaking in 2030 would be challenging given the expected economic growth rates for at least the next two decades.

Although the Indian steel industry and the government are aiming for a reduction in $CO_2$ emissions as well as in specific energy consumption, realization demands stronger commitment in the development of low-carbon and carbon-neutral technologies and the rapid implementation of best available technologies to improve the current processes. Major technical breakthroughs are needed as well as research promotion in the field of carbon-lean/carbon-free iron making. It can be achieved through enhanced hydrogen utilization in the process, which is linked to the production and storage for hydrogen. In addition, the growing supply of scrap is a strong trend in India, which means a growing demand on scrap-based processes, which correspondingly increases the demand for carbon-neutral electricity.

## 6. Conclusions

The current state of the Indian steel industry and future visions for the year 2030 and beyond were examined with the main focus on the mitigation of $CO_2$ emissions in relation to the global trends, goals, and agreements. The authors came to the following conclusions:

(1) As a developing country, India has set an ambitious target to more than double its steel production up to 250 MT in 2030. A significant share of the growth is based on BF ironmaking and coal-based DRI production, in other words, utilizing fossil coal as the primary energy source. This makes the $CO_2$ challenge extremely hard.

(2) A positive matter is that natural gas will partly substitute for coal both in BF ironmaking and direct reduction. It will generate less $CO_2$ emissions, but eventual methane escape should be strictly eliminated.

(3) Owing to these circumstances, it is evident that the $CO_2$ emissions will continue growing, and the peaking year for the steel industry's emissions is escaping far ahead. According to the plans of the government and the steel industry, the turning point is expected shortly after 2030.

(4) The current levels of energy consumption and $CO_2$ emissions in the Indian steel industry are much higher than the world average due to weaknesses in raw materials and energy as well as technological deficiencies. By applying the best available technologies in retrofitting plants and in new constructions, it is realistic to cut energy consumption and emission levels by 35–40% from the present levels toward the end

of the 2020s. This will greatly decelerate the emissions' rise owing to the growing production, but it will not stop it.

(5) To get on a declining track with $CO_2$ emissions, stronger decarbonization means are mandatory. A considerable share of new steel plant investments should be based on hydrogen reduction and green electricity in all operations, including hydrogen production. This is the way to reach the emissions' peak and turn downward toward carbon-neutral steelmaking in the middle of the century. By strong commitment to carbon-neutral technologies in new investments, the Indian steel industry can take a forerunner position in fighting climate warming.

**Author Contributions:** S.P.S.—Literature survey, analysis, plotting the figures and drafting the manuscript. V.N.N.—Conceptualization of the paper, analysis and review of the manuscript. S.M.—Conceptualization of the paper, analysis and review of the manuscript. S.C.—Conceptualization of the paper, analysis and review of the manuscript. L.E.K.H.—Process development and energy inspections—critical evaluation, briefing in writing, validation. All authors contributed significantly. All authors have read and agreed to the published version of the manuscript.

**Funding:** This research received no external funding.

**Institutional Review Board Statement:** Not applicable.

**Informed Consent Statement:** Not applicable.

**Conflicts of Interest:** The authors declare no conflict of interest.

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
