# Peer review of "Challenges and Outlines of Steelmaking toward the Year 2030 and Beyond—Indian Perspective"

_metals, doi:10.3390/met11101654_

Round 1

Reviewer 1 Report

Paper presents interesting overview of the current state and the future of Indian steel sector. Paper is well written but there are few remarks as follows:

1)  statement on p. 6 about the effect caused by DRI carbon on EAF operation shall be reviewed: if carbon is present in DRI and is bound to cementite it may increase power efficiency of EAF thanks to faster melting.

2) Why HIsmelt and Finex (both suitable for low grade ore and coal, and more mature compared to HIsarna) are not considered as new technology candidates?

3) On p. 7 BAT for BF-BOF route is 16.4 GJ/TCS but Fig 9 refers 16.7. Moreover, it is unclear how reaching of the world level BAT can be possible under specific conditions of Indian raw materials and whether targets set in Fig 9 are realistic indeed. Although data presented in Fig 9 are not obtained by authors, some comments shall be provided on this matter.

4) Question mark seems strange to appear in the Conclusions.

Author Response

Reviewer 1 Comments

Paper presents interesting overview of the current state and the future of Indian steel sector. Paper is well written but there are few remarks as follows:

Response – Authors thank the reviewer for the comments.

Comment 1:  Statement on p. 6 about the effect caused by DRI carbon on EAF operation shall be reviewed: if carbon is present in DRI and is bound to cementite it may increase power efficiency of EAF thanks to faster melting.

Response -

Authors welcome the reviewer’s suggestions.  An additional sentence has been incorporated  regarding EAF in the manuscript as follows.

However, manufacturers who use EAF shall prefer DRI from gas based units as higher carbon content gives higher energy efficiency[21].

Comment 2: Why HIsmelt and Finex (both suitable for low grade ore and coal, and more mature compared to HIsarna) are not considered as new technology candidates?

Response:

Authors agree with the views of the reviewer on this issue. We have included HIsarna alone as a pilot plant was about to get installed by Tata Steel in India.  A separate paragraph on Finex Technology has also been added in the manuscript.

Finex Technology - Considering the large amount of iron ore fines produced during mining as well as the abundance of non-coking coal (combined with limited resources of coking coal), Finex technology was considered to be one of the potential technologies for India [36].  In 2015, POSCO was planning to install 12MT steel plant using Finex technology [37]. However, because of non-technical reasons the plan was discontinued. Considering the success of COREX technology in two plants in India, Finex technology is a viable route for the future of Indian steel sector.

Suitable references are also given.

Comment 3: On p. 7 BAT for BF-BOF route is 16.4 GJ/TCS but Fig 9 refers 16.7. Moreover, it is unclear how reaching of the world level BAT can be possible under specific conditions of Indian raw materials and whether targets set in Fig 9 are realistic indeed. Although data presented in Fig 9 are not obtained by authors, some comments shall be provided on this matter.

Response

We agree reviewer’s comments on reaching the world level BAT. We have provided following comments for explaining our views further.

Although there exists a substantial potential to save energy by adopting best practices and newest innovations for reducing energy consumption, reaching this target can be quite challenging considering the quality of raw materials in India. Thus, the authors opine that arriving a target specific energy consumption considering the local raw material quality can be quite fruitful in defining the road map for steel technology for India

4) Question mark seems strange to appear in the Conclusions. ]

Response -

Respective statement having question mark is removed

Reviewer 2 Report

This manuscript is well written, and gives a comprehensive assessment of the future paths for the steel industry in INDIA, to grow in a sustainable fashion. However, given that steel production is planned to be double that of today, in 2050, but will still based on carbon, they show how the improved plant efficiencies can be significantly improved to almost "hold the line", on net CO2 emissions, with double the steel production in 2050, compared to 2020 steel output figures. The real solution is to move to a hydrogen based reduction of Fe2O3, but they do not elaborate on how INDIA might achieve this output over the longer term. I think further analysis of CO2 sequestration  with HISARNA steelmaking, is warranted, plus further details of the prospects for green hydrogen production, in INDIA, would improve the value of this paper. Nevertheless, the strong point of the m/s, is the comprehensive evaluation of the metallurgical field available to them, going forwards to 2050. 

Author Response

Reviewer 2 comments

This manuscript is well written, and gives a comprehensive assessment of the future paths for the steel industry in INDIA, to grow in a sustainable fashion. However, given that steel production is planned to be double that of today, in 2050, but will still be based on carbon, they show how the improved plant efficiencies can be significantly improved to almost "hold the line", on net CO2 emissions, with double the steel production in 2050, compared to 2020 steel output figures. The real solution is to move to a hydrogen-based reduction of Fe2O3, but they do not elaborate on how INDIA might achieve this output over the longer term. I think further analysis of CO2 sequestration with HISARNA steelmaking, is warranted, plus further details of the prospects for green hydrogen production, in INDIA, would improve the value of this paper. Nevertheless, the strong point of the m/s, is the comprehensive evaluation of the metallurgical field available to them, going forwards to 2050. 

Response:

Authors thank the reviewer for the very constructive comments.  Based on the reviewer’s comments we have added a few paragraphs on carbon capture. 

One of the best ways to capture CO2 is the natural sequestration by plants. India aims to create a sink of 2.5 – 3 billion tons of CO2 through additional forestation and tree cover by 2030. Apparently, through large level community involvement, India has launched Mahatma Gandhi National Rural Employment Guarantee Act (MGNREGA) which focuses on environment and natural resource conservation. A detailed review of its potential impact on carbon sequestration through conservation of green natural resources has been carried out by Indian Institute of Science, Bangalore [43].  Steel plants are also actively involved in such afforestation drives [44].

Transition to processes based on low-carbon and carbon-free clean energy is a key factor to conserve the energy and environment. During the transition period Carbon Capture and Storage (CCS) affords further means to cut CO2 emissions. With CCS technology, upto 90% of the CO2 emitted can be captured, compressed at high pressure, converted into a liquid and injected to geological formation sites to be stored underground without significant leak for hundreds of years. Implementation of this process along with TGR-OBF, HIsarna etc. can significantly reduce the emissions levels. One major concern for CCS deployment in India is to find accurate and suitable geological site for installation of CCS. Another issue is that CCS significantly increases the cost of electricity while reducing net power output which is often cited as being the biggest barrier to acceptability of CCS in India[45].

As mentioned earlier CO2 can be injected into geological formation sites such as saline aquifers, basalt formations, depleted oil and gas fields and non-minable coal seams to fixate CO2 as carbonates.  An initial geological study suggests that 500 – 1000 Gt of CO2 can be potentially stored around the sub continent [46]. More specific studies pertaining to India in this direction are needed and further, India is closely studying these developments in developed countries in terms of implementation in the actual scale and subsequently planning to adopt them

Also,

  • We have changed the title of sec 4.3 as Carbon Sinks, Capture and Storage

Reviewer 3 Report

The manuscript "Challenges and Prospects of Steelmaking towards the Year 2030 and beyond - Indian Perspective" is categorized as a review paper to evaluate the issues around the steel industry in India. The main proposed achievement is "outcomes of this study will help to peak the CO2 emissions and turn downward towards carbon-neutral production". I couldn't get this message as a review paper. Most statements are mentioned without referencing valid articles, books, or scientific reports (e.g., subsection 3.2: Processes).

 The title of this review is not compatible with its content and should be thoroughly reconsidered in terms of structure, syntax, content, references, and abbreviations. For instance about abbreviations: Direct Reduced Iron (DRI) or Hot Briquetted Iron (HBI) should be "direct reduced iron (DRI) or hot briquetted iron (HBI)". A table of content is necessary for any review paper.

Author Response

Reviewer 3 comments

Comment:

The manuscript "Challenges and Prospects of Steelmaking towards the Year 2030 and beyond - Indian Perspective" is categorized as a review paper to evaluate the issues around the steel industry in India. The main proposed achievement is "outcomes of this study will help to peak the CO2 emissions and turn downward towards carbon-neutral production". I couldn't get this message as a review paper. Most statements are mentioned without referencing valid articles, books, or scientific reports (e.g., subsection 3.2: Processes).

The title of this review is not compatible with its content and should be thoroughly reconsidered in terms of structure, syntax, content, references, and abbreviations. For instance, about abbreviations: Direct Reduced Iron (DRI) or Hot Briquetted Iron (HBI) should be "direct reduced iron (DRI) or hot briquetted iron (HBI)". A table of content is necessary for any review paper.

Response from authors and corrections made in the manuscript:

Authors agree for your valuable suggestions and improvement points and accordingly following revisions are made in the manuscript

  • Title of the manuscript is changed from “Challenges and prospects of steelmaking towards the year 2030 and beyond – Indian perspective” to “Challenges and outlines of steelmaking towards the year 2030 and beyond – Indian perspective”
  • we have added some more references in section 3.2 and further, additional things have been added regarding CCS giving the geological potential of India for CCS, studies on steel plant slag to be used for CCS and also increasing the greenery cover. 
  • Corrections were made regarding the usage of abbreviations all along the manuscript.

Round 2

Reviewer 3 Report

The revision has not been improved at all. Refer to the previous comment: "The manuscript "Challenges and Prospects of Steelmaking towards the Year 2030 and beyond - Indian Perspective" is categorized as a review paper to evaluate the issues around the steel industry in India. The main proposed achievement is "outcomes of this study will help to peak the CO2 emissions and turn downward towards carbon-neutral production". I couldn't get this message as a review paper. Most statements are mentioned without referencing valid articles, books, or scientific reports".